# Perturbation of Copper Homeostasis Sensitizes Cancer Cells to Elevated Temperature

**DOI:** 10.3390/ijms25010423

**Published:** 2023-12-28

**Authors:** Enzo M. Scutigliani, Jons van Hattum, Fernando Lobo-Cerna, Joanne Kruyswijk, Maja Myrcha, Frederique E. G. A. Dekkers, Ron A. Hoebe, Finn Edwards, Jetta J. Oppelaar, Liffert Vogt, Sanne Bootsma, Maarten F. Bijlsma, Daisy I. Picavet, Johannes Crezee, Jorg R. Oddens, Theo M. de Reijke, Przemek M. Krawczyk

**Affiliations:** 1Department of Medical Biology, Amsterdam UMC Location University of Amsterdam, Meibergdreef 9, 1105 AZ Amsterdam, The Netherlands; e.m.scutigliani@amsterdamumc.nl (E.M.S.); f.h.lobocerna@amsterdamumc.nl (F.L.-C.); joanne.kruyswijk@student.uva.nl (J.K.); m.myrcha@amsterdamumc.nl (M.M.); f.e.g.a.dekkers@amsterdamumc.nl (F.E.G.A.D.); f.l.edwards@amsterdamumc.nl (F.E.); d.i.picavet@amsterdamumc.nl (D.I.P.); 2Cancer Center Amsterdam, Treatment and Quality of Life, Cancer Biology and Immunology, De Boelelaan 1118, 1081 HV Amsterdam, The Netherlands; j.w.vanhattum@amsterdamumc.nl (J.v.H.); h.crezee@amsterdamumc.nl (J.C.); j.r.oddens@amsterdamumc.nl (J.R.O.); t.m.dereyke@amsterdamumc.nl (T.M.d.R.); 3Department of Urology, Amsterdam UMC Location University of Amsterdam, Meibergdreef 9, 1105 AZ Amsterdam, The Netherlands; 4Department of Internal Medicine, Section of Nephrology, Amsterdam UMC Location University of Amsterdam, Meibergdreef 9, 1105 AZ Amsterdam, The Netherlands; j.j.oppelaar@amsterdamumc.nl (J.J.O.); l.vogt@amsterdamumc.nl (L.V.); 5Amsterdam Cardiovascular Sciences, Microcirculation, Meibergdreef 9, 1105 AZ Amsterdam, The Netherlands; 6Center for Experimental and Molecular Medicine, Laboratory of Experimental Oncology and Radiobiology, Amsterdam UMC Location University of Amsterdam, Meibergdreef 9, 1105 AZ Amsterdam, The Netherlands; s.j.bootsma@amsterdamumc.nl (S.B.); m.f.bijlsma@amsterdamumc.nl (M.F.B.); 7Cancer Center Amsterdam, Cancer Biology, De Boelelaan 1118, 1081 HV Amsterdam, The Netherlands; 8Oncode Institute, Jaarbeursplein 6, 3521 AL Utrecht, The Netherlands; 9Department of Radiation Oncology, Amsterdam UMC Location University of Amsterdam, Meibergdreef 9, 1105 AZ Amsterdam, The Netherlands

**Keywords:** hyperthermia, heat stress, elesclomol, cancer, copper

## Abstract

Temporary elevation of tumor temperature, also known as hyperthermia, is a safe and well-tolerated treatment modality. The efficacy of hyperthermia can be improved by efficient thermosensitizers, and various candidate drugs, including inhibitors of the heat stress response, have been explored in vitro and in animal models, but clinically relevant thermosensitizers are lacking. Here, we employ unbiased in silico approaches to uncover new mechanisms and compounds that could be leveraged to increase the thermosensitivity of cancer cells. We then focus on elesclomol, a well-performing compound, which amplifies cell killing by hyperthermia by 5- to 20-fold in cell lines and outperforms clinically applied chemotherapy when combined with hyperthermia in vitro. Surprisingly, our findings suggest that the thermosensitizing effects of elesclomol are independent of its previously reported modes of action but depend on copper shuttling. Importantly, we show that, like elesclomol, multiple other copper shuttlers can thermosensitize, suggesting that disturbing copper homeostasis could be a general strategy for improving the efficacy of hyperthermia.

## 1. Introduction

Cancer remains a major health issue, with approximately 3.1 million new cases and 1.9 million deaths in Europe in 2018 [1], creating an urgent need for more effective treatment strategies. Mild hyperthermia (i.e., elevation of tumor temperature to 40–43 °C for 1–2 h), hereafter referred to as hyperthermia, has a demonstrable clinical track record in the treatment of various tumor types, including breast, head and neck, rectum, bladder, lung, and cervix cancers [2,3,4,5,6,7,8]. 

Hyperthermia has wide-ranging effects on tissues and cells—it enhances tissue temperature, perfusion, and oxygenation, thereby (i) boosting radiotherapy by increasing oxygen concentrations [6,9,10,11,12,13,14,15,16], (ii) promoting drug delivery, especially when used in combination with thermosensitive delivery systems [17,18,19,20], (iii) triggering immunostimulatory effects [21,22], and (iv) battling against adverse microenvironmental conditions that support disease progression and tumor metastasis, including hypoxia and nutrient depletion [23,24,25]. On the cellular level, hyperthermia affects virtually all organelles by inducing proteotoxic stress and altering membrane characteristics [26], and the transient impairment of DNA repair pathways improves radio- and chemotherapy [27]. 

Despite hyperthermia’s clinical successes, it remains technically challenging to apply, and its effects can be effectively counteracted by cancer cells through a number of mechanisms, collectively known as the heat stress response [26,28,29]. Apart from the fact that the heat stress response allows cancer cells to survive lethal heat insults, it also contributes to the development of tumoral thermotolerance—a state of enhanced resistance to heat stress. After its activation, this state can last in the order of days, thereby limiting the frequency at which hyperthermia can be supplemented to a treatment schedule [29]. New strategies to thermosensitize cancer cells, either through targeted inhibition of the heat stress response or through effective manipulation of other cellular functions, would likely enhance its clinical efficacy.

In this study, we attempt to uncover new thermosensitization strategies by employing several in silico approaches that link thermosensitivity to the thermal stability of proteins [30], as well as to the transcriptional profiles and gene dependencies of cancer cells. Informed by the results of these analyses, we construct a panel of drugs that target mechanisms that are presumably involved in cellular responses to heat stress, and we identify the copper ionophore elesclomol as a novel thermosensitizer. We then show, in vitro, that brief exposure to elesclomol and hyperthermia outperforms the current clinical standard treatments in bladder cancer cells, and we explore the underlying mode of action. Interestingly, our results suggest that elesclomol thermosensitizes by increasing the intracellular copper concentration, but independent of its many known mechanisms of action, including global activation of cellular stress responses, induction of reactive oxygen species (ROS), inhibition of oxidative phosphorylation (OXPHOS), and promotion of cuproptosis. We also show that multiple other copper shuttlers are efficient thermosensitizers, suggesting that disturbing copper homeostasis could be a general strategy for boosting the efficacy of hyperthermia.

## 2. Results 

### 2.1. In Silico Analyses Identify Cellular Energy Metabolism and ROS Homeostasis as Potential Targets for Thermosensitization

To uncover cellular processes that can be targeted to induce thermosensitization, we hypothesized that processes important for surviving heat stress would involve thermostable proteins, and we explored this hypothesis using the human protein Meltome dataset [30]. We marked proteins with melting temperatures in the upper and lower quartiles of the data as thermostable and -labile, respectively (Figure 1A), and performed a gene overrepresentation analysis for Gene Ontology (GO) terms and hallmark signatures for both categories (Figure 1B,C) [31,32]. Among the thermostable proteins, we found terms related to the chaperone network, protein degradation pathways [26], redox reactions, ROS scavenging, OXPHOS, glycolysis, fatty acid oxidation, and the tricarboxylic acid (TCA) cycle, among others. GO terms enriched for thermolabile proteins were often related to processes known to be impaired by heat, including DNA repair, cytoskeletal functions, transcription, and translation [26,27,33].

In another unbiased approach, we investigated which genes likely contribute to cellular heat stress responses. To this end, we first determined the thermosensitivity of a panel of 15 cancer cell lines, and we correlated their thermosensitivity with global RNA expression levels and with CRISPR-inferred gene dependency using the DepMap Consortium dataset (Figure 1D–H). Thermoresistance was positively correlated with the expression of genes involved in the unfolded protein response, mTORC, MYC signaling [34,35,36], and OXPHOS (Figure 1G), as well as with the dependency on complex I of the OXPHOS chain (Figure 1H). On the other hand, processes known to promote cell death upon heat exposure, including interferon activation [37,38,39], were negatively correlated. 

In conclusion, different unbiased analyses suggested a number of potential new strategies to thermosensitize cancer cells, in particular inhibiting thermoresistant processes (i.e., OXPHOS) or amplifying the negative effects of heat on thermolabile processes (i.e., interferon activation).

### 2.2. Targeting Energy Metabolism and ROS Homeostasis as a Thermosensitization Strategy

Next, we decided to focus on energy production and ROS homeostasis, which, based on our unbiased analyses as well as the available literature, appear to play a role in thermoresistance [26,40,41,42]. First, we checked whether interfering with cellular energy production can thermosensitize cancer cells. We found that total ATP levels were not significantly affected by hyperthermia (Figure 2A), whereas mitochondrial activity, quantified by mitochondrial membrane potential, was disturbed in some of the tested cell lines early after the treatment (Figure 2B,C), but these changes were not correlated with thermosensitivity (Figure 2D). Shifting the cellular energy production to OXPHOS by cultivation in glucose- and galactose-rich media did not cause thermosensitization either (Appendix A), even though OXPHOS confers thermoresistance [41] and may provide backup when mitochondrial ATP output is insufficient. 

To evaluate whether cellular ROS levels contribute to thermotoxicity, we measured general ROS and mitochondrial superoxide levels. Although general ROS levels remained unaltered after hyperthermia, superoxide levels increased (Figure 2E,F). ROS scavengers did not induce thermoresistance, however (Figure 2G), arguing against a major role for ROS production in hyperthermia-induced cell death.

We then tested a set of 11 compounds that target the aforementioned pathways (Table 1, Figure 2I), using the thermal enhancement ratio (TER—ratio between cell viability after normothermia and hyperthermia) as a readout. We evaluated the thermosensitizing capacity of these compounds in a set of cervix and bladder cancer cell lines, as corresponding tumor types are routinely treated with hyperthermia in clinical settings. Inhibitors targeting fatty acid oxidation, glycolysis, OXPHOS, and glutamine oxidation failed to cause thermosensitization (Figure 2H and Appendix A), likely reflecting the metabolic flexibility of cancer cells under hyperthermic conditions. Interestingly, the ROS-inducing agents menadione and hydrogen peroxide were excellent thermosensitizers at micromolar-range concentrations (Figure 2H), and exposure to ROS scavengers prevented thermosensitization by menadione (Appendix A) [43]. In addition, the copper ionophore disulfiram and the ROS-inducing copper ionophore elesclomol thermosensitized most cell lines at relatively low, sub-micromolar concentrations (Figure 2H) [44,45].

### 2.3. Elesclomol Is a Potent Thermosensitizer

Since elesclomol thermosensitized all tested cell lines and is a clinically advanced drug candidate [46,47], we decided to further explore its potential in the context of hyperthermia. Even though elesclomol was initially identified as a potent ROS inducer (Figure 3A) [48,49], follow-up studies demonstrated that it shuttles copper from the extracellular environment to (predominantly) mitochondria [50], triggering copper reduction that releases free radicals in a Fenton reaction [51,52]. Aside from ROS generation, elesclomol interferes with mitochondrial energy production by uncoupling OXPHOS [53]. Recent studies demonstrated that elesclomol induces a unique form of copper-dependent cell death called cuproptosis by triggering aggregation of TCA-cycle proteins through an interaction with ferredoxin 1 (FDX1) [54,55]. Elesclomol has a short half-life when applied systemically [46,47], so we decided to further study its thermosensitizing potential using bladder cancer cell lines, as bladder chemohyperthermia (CHT) can easily achieve high local drug concentrations.

When establishing the optimal elesclomol/copper ratio, we found that a relatively high concentration of either can increase TER, resulting in 5–20-fold sensitization to hyperthermia (Figure 3B and Appendix A). To compare the thermosensitizing potential of elesclomol to that of existing bladder CHT combination therapies, we juxtaposed it with mitomycin C, epirubicin, and gemcitabine. Elesclomol generated the highest TER of the tested compounds (Figure 3C,D), suggesting that it could be considered for bladder CHT.

### 2.4. Elesclomol and Hyperthermia Induce Alterations in Proteostasis and the Transcriptome

To establish how elesclomol thermosensitizes, we first explored a scenario in which induction of global (oxidative) stress would amplify the effects of hyperthermia on the cellular proteostasis network. A pulsed treatment with combinations of hyperthermia and elesclomol, however, induced neither upregulation of the major stress marker HSP70 [48,56] (Appendix A) nor significant changes in global ubiquitination patterns—a surrogate for recovery from proteome stress [57] (Appendix A).

Next, we evaluated the effects of hyperthermia and elesclomol on the transcriptome of the bladder cancer cell line T24. Interestingly, combined treatment did not lead to unique expression alterations in processes known to be affected by elesclomol (i.e., OXPHOS, ROS homeostasis, and cuproptosis) (Figure 4A–D) or hyperthermia (i.e., DNA repair; Figure 4E–G). We also found that several cellular processes were enriched for genes that are uniquely differentially expressed after combination treatment (Appendix A), and a leading edge analysis (Appendix A) identified a set of genes that drive the enrichment of processes related to non-coding RNA and tRNA, albeit these effects appeared to be shared between the combination therapy and hyperthermia alone (Appendix A). Therefore, we can conclude that combining elesclomol and hyperthermia causes widespread changes in gene expression, but no particular process can be pinpointed as a driver of elesclomol-mediated thermosensitization.

### 2.5. The Thermosensitizing Effects of Elesclomol Are Not Caused by Interference with Energy or ROS Production

To evaluate whether hyperthermia modulates elesclomol’s inhibitory effects on OXPHOS [53], we quantified changes in mitochondrial membrane potential but found no consistent effects (Figure 5A). Surprisingly, and in contrast to earlier reports, the total ATP levels increased in most cell lines immediately after the treatments (Figure 5B). It is known, however, that ATP levels increase prior to cell death [59], and since we also found a significant negative correlation between cell survival and ATP levels (Appendix A), our measurements are likely to be confounded by these effects. Combined with the observation that OXPHOS inhibition does not thermosensitize (Figure 2H), we can conclude that if elesclomol inhibits OXPHOS and ATP production at all, inactivation of these processes does not significantly contribute to thermosensitization.

The consensus that elesclomol mainly exerts toxic effects by generating ROS [50,51], combined with our finding that elevated ROS levels sensitize cells to hyperthermia (Figure 2H and Appendix A), prompted us to question the contribution of ROS to elesclomol-induced thermosensitization. We found no changes in overall ROS after hyperthermia, and simultaneous treatment with elesclomol failed to increase superoxide levels, even at elesclomol concentrations that led to profound thermosensitization (Figure 5C,D and Appendix A). Additionally, even though NAC was able to partially rescue cell viability in the lower range of tested elesclomol concentrations, no rescue was observed under other conditions (Figure 5E). Although increased ROS levels caused by elesclomol have been shown to trigger a cell death mechanism known as ferroptosis [60], we could not rescue cells from elesclomol-related toxicity or thermosensitization with the use of ferroptosis inhibitors (Appendix A). Together, these results suggest that enhanced ROS levels, OXPHOS inhibition, and ferroptosis are not the main drivers of thermosensitization.

### 2.6. Cuproptosis Does Not Drive Thermosensitization by Elesclomol

Subsequently, we investigated whether hyperthermia can stimulate cuproptosis—copper-dependent cell death reportedly triggered by elesclomol [55]. One mechanism could be enhanced copper shuttling due to the alteration of membrane characteristics, similar to the enhanced uptake of platinum-based compounds after hyperthermia [61,62]. However, hyperthermia did not elevate intracellular copper levels (Figure 6A,B). We also evaluated the thermosensitivity of a panel of cell lines that are deficient in FDX1-2, FDXH, and LIAS—key players in the cuproptosis pathways [55]—and found that knockout ABC1 cell lines displayed sensitivity similar to the wildtype and CRISPR-control cell lines (Figure 6C). However, although we could confirm that FDX1 and FDX2 modify elesclomol’s toxicity (Appendix A), ABC1 cells could not be thermosensitized with the compound, even at relatively high concentrations (Figure 6D). We also quantified the aggregation of DLAT, a protein involved in the TCA cycle that was shown to aggregate upon elesclomol exposure [55]. Although there was some tendency towards an increase in the number of foci with elesclomol (Figure 6E,F), hyperthermia did not enhance this further. Combined, these results suggest that cuproptosis is not the mechanism underlying elesclomol’s thermosensitizing capacity.

### 2.7. Molecules Interfering with Copper Homeostasis Generally Thermosensitize

Finally, given that copper plays an essential role in the toxicity of elesclomol, we questioned whether copper-binding agents or elevated intracellular copper levels generally drive thermosensitization. Interestingly, even relatively high environmental copper levels did not sensitize cells to hyperthermia (Figure 7A), and limiting its bioavailability by pretreating cells with the copper chelator TTM did not confer thermoprotection (Figure 7B). In contrast, pretreatment with TTM did rescue cells exposed to elesclomol and completely abolished its thermosensitizing effects (Figure 7C). Importantly, when we subjected T24 bladder cancer cells to a panel of copper-binding agents (i.e., zinc pyrithione, thiram, tetramethylthiuram) and copper ionophores (i.e., elesclomol, GTSM-Cu, 8HQ, clioquinol), we observed that copper ionophores thermosensitized to comparable or even higher extents than elesclomol (Figure 7D). Interestingly, in contrast to elesclomol and GTSM-Cu, copper was not required for thermosensitization by the copper ionophores 8HQ and clioquinol, as the viability of cells exposed to these compounds was not rescued by pretreatment with TTM (Figure 7E). These findings unequivocally demonstrate that thermosensitization by elesclomol is mediated by copper, and that interfering with copper homeostasis is generally thermosensitizing.

## 3. Discussion

In this study, we attempted to uncover and exploit novel pathways that play important roles during cellular responses to heat stress, with the goal of stimulating the development and improvement of hyperthermia-based cancer treatments. Through various in silico analyses, we identified multiple processes potentially affected by heat (Figure 1). Some of them, including the proteasomal degradation system, mTORC and MYC signaling [26,34,35,36], DNA repair, cytoskeletal functions, transcription, and translation [26,27,33], were already known to be impacted by heat or implicated in heat stress responses. Therefore, we focused on the energy generation and ROS metabolism, which have been less studied in the context of hyperthermia, and which are clinically targetable. 

The (combined) manipulation of primary catabolic pathways failed to thermosensitize cells (Figure 2H and Appendix A), underscoring the metabolic flexibility of cancer cells under heat stress. Interestingly, inducing an ROS overload pharmacologically with menadione or hydrogen peroxide did thermosensitize cells, but not with PQ and DMNQ. Therefore, further experimentation is required to investigate why thermosensitization is not a shared feature of ROS inducers and might be governed by other modes of action; menadione, for instance, has also been shown to be an effective inhibitor of DNA polymerase γ [43,63]. From our panel of compounds, elesclomol—a drug with proven clinical tolerability [46,47,64] that interferes with both energy production and ROS—appeared to consistently thermosensitize different cell lines and was chosen for further evaluation (Figure 2). In terms of in vitro performance, elesclomol outperformed conventional CHT for the treatment of bladder cancer, especially when used at an optimal ratio with copper (Figure 3), but parallel ongoing efforts in our laboratory are being directed towards exploring the efficacy and feasibility of the vitamin K2 precursor menadione for hyperthermia-based treatments [65,66]. 

Our targeted attempts to pinpoint the mechanism(s) driving the thermosensitizing properties of elesclomol ruled out a number of possibilities: neither the cellular proteostasis network (Appendix A), induction of ROS, uncoupling of OXPHOS, or cuproptosis (Figure 5 and Figure 6) seem likely candidates. Interestingly, and contrary to previous research [51,52,53,55], we also did not consistently observe the abovementioned alterations after elesclomol monotherapy. Differences in experimental procedures might provide an explanation—while we used a clinically relevant one-hour pulsed exposure, the observation that elesclomol uncouples OXPHOS is based on isolated mitochondria treated with highly toxic concentrations [53], and the induction of ROS has been observed within minutes after the end of a prolonged exposure [51,52]. While copper is essential for thermosensitization (Figure 3B), we could not find indications that cuproptosis is involved in elesclomol-mediated thermosensitization—hyperthermia neither elevated intracellular copper levels nor promoted the subsequent aggregation of the TCA cycle protein DLAT [55] (Figure 7). Again, a possible explanation is that the exposure to elesclomol was too short to induce cuproptosis, as suggested by the more profound effect of an FDX1 knockout on elesclomol sensitivity after a longer exposure (Figure 7E).

Which effects drive elesclomol-mediated thermosensitization? Our unbiased efforts to answer this question by transcriptomic profiling of T24 bladder cancer cells exposed to elesclomol and hyperthermia revealed a number of patterns in gene expression changes, but a particular process that could explain the thermosensitizing effects failed to emerge (Figure 4). The mechanism of action is clearly dependent on copper, however (Figure 3B and Figure 7C), and thermosensitization appears to be shared among copper ionophores (Figure 7). As recent data show that copper shuttling by elesclomol and other copper ionophores can trigger copper-related toxicities in a cuproptosis-independent fashion in other cellular compartments [67], future studies should explore which cellular components and processes are affected by copper, as well as whether these processes are involved in the thermosensitizing phenotype of elesclomol. 

Our data on elesclomol pointed the way towards a much more general finding that the molecules interfering with copper homeostasis are generally thermosensitizing. Intriguingly, this includes both copper chelators and ionophores (Figure 7D), and it is not always prevented by the copper chelator TTM (Figure 7E). The latter results could be explained by the different copper affinities of these compounds (which could be higher than the affinity of TTM) or, alternatively, by additional copper-independent mechanisms of action for 8HQ [68] and clioquinol [69]. In any case, it will be instrumental to explore how exactly copper homeostasis is related to cellular responses to heat, and how this relationship could be exploited in cancer treatment.

What steps should be taken to investigate the potential application of elesclomol as a thermosensitizer? Elesclomol has been successfully tested in phase II/III trials, but its short half-life is thought to have played a key negative role in the results of late-stage trials [46,47]. Combined with our finding that optimal thermosensitization is achieved by mixed formulations of elesclomol and copper, a local, short therapeutic exposure at high concentrations would therefore be the most practical route to further optimize this novel therapy and circumvent adverse copper-related systemic effects. Three hyperthermia-based treatments match these requirements: the already-discussed bladder CHT, hyperthermic intraperitoneal chemotherapy (HIPEC)—used for the treatment of peritoneal metastases of gastrointestinal, ovarian, and other tumors [70,71]—and hyperthermic isolated limb perfusion (HILP) [72,73]. Animal experiments that assess the efficacy of elesclomol, potentially in combination with varying copper ratios, in a bladder CHT setting [74] could pave the way for a phase I study exploring therapy-related toxicities in the short-term. Combined with more mechanistic studies, these insights will shed light on the potential of elesclomol to improve hyperthermia-based treatments in clinical cancer care.

## 4. Materials and Methods

### 4.1. Cell Culture

All cells were cultured at 37 °C and 5% CO_2_. SiHa, HeLa, J82, FADU, C33A, and CASKI cells were maintained in EMEM (Lonza, Basel, Switzerland), while T24, RT112, EGI1, TFK1, A375, T47D, and MCF7 cells were maintained in DMEM (Gibco, Grand island, NY, USA), and RCM1, HT55, and HUTU80 cells were maintained in DMEM/F12 (Gibco), with 10% FBS (Gibco), 100 U/mL penicillin (Gibco), 100 U/mL streptomycin (Gibco), and 2 mM L-glutamine (Gibco), unless stated otherwise. For experiments involving adjusted levels of glucose and galactose, cells were cultured in DMEM lacking glucose, L-glutamine, phenol red, sodium pyruvate, and sodium bicarbonate with 10% dialyzed serum, penicillin, streptomycin, 1 mM sodium pyruvate, and 10 mM of either glucose or galactose (all from Merck, Darmstadt, Germany) for 24 h before the start of treatment.

### 4.2. Hyperthermia, Drug Treatments, and Reagents

Hyperthermia was applied in a 42 °C water bath at 5% CO_2_ for one hour, unless specified otherwise. N-acetyl-l-cysteine, Trolox, DMSO, MitoQ, and tetrathiomolybdate (all from Merck) were present from 6 h before until 24 h after the hyperthermia treatment. Elesclomol (Selleck, Houston, TX, USA), epirubicin hydrochloride, mitomycin C, gemcitabine hydrochloride, menadione, and hydrogen peroxide (all from Merck) were added 5 min prior to hyperthermia and removed 5 min afterwards. Thiram, TMT, Zn-pyrithione (Sigma, St. Louis, MO, USA), 8HQ, clioquinol, and GTSM-Cu (Medchem Express, Monmouth Junction, NJ, USA) were added 30 min prior to hyperthermia and removed 24 h after. Elesclomol was used in combination with copper(I) chloride (Merck) at an equimolar ratio unless stated otherwise. In cell viability experiments involving mono-exposure to copper, copper(I) chloride was added 5 min before hyperthermia and left until the end of the experiment. Antimycin-A (Selleck) and FCCP (Abcam, Cambridge, UK) were added 1 h prior to hyperthermia and left until the end of the experiment. DMNQ (Medchem Express) and paraquat dichloride (Merck) were added 5 min prior to hyperthermia and left until the end of the experiment. BPTES, etomoxir, and UK5099 (all from Selleck) were added 4–6 h prior to hyperthermia and removed after 24 h.

### 4.3. Cell Viability Assays

Cells were treated in flat-bottomed 24- or 96-well plates (Greiner, Kremsmünster, Austria). At 80–90% confluence of the untreated control, viability was assessed using PrestoBlue™ (Thermo Fisher, Waltham, MA, USA) according to the manufacturer’s guidelines, using a Clariostar plate reader (BMG Labtech, Ortenberg, Germany).

### 4.4. In Silico Analyses

Gene set enrichment and overrepresentation analyses were performed in R with clusterProfiler [75], using signatures from the msigdbr package. Heatmap visualizations were generated by the pheatmap package. For the prediction of thermostable and thermosensitive pathways, the Meltome dataset by Jarzab et al. [30] was used. For correlation analyses, we used the DepMap (https://depmap.org/, accessed on 23 February 2022) RNA expression dataset 22Q2. CRISPR dependency scores, as determined by [76], were retrieved from dataset 21Q4. Gene set enrichment analysis was performed on a gene list sorted by Pearson correlation strength. The correlation analysis based on RNA expression was performed on the cell lines A375, C33A, CASKI, EGI1, FADU, HELA, HT55, HUTU80, MCF7, RCM1, RT112, SIHA, T24, T47D, and TFK1. Correlation analysis based on CRISPR dependency was performed using the same panel minus HUTU80 and HELA.

### 4.5. Quantification of Total ROS and Mitochondrial Superoxide

For total ROS quantification, 10,000 cells/well were seeded in flat-bottomed 96-well plates, incubated with CellROX™ Green (Thermo Fisher), and fixed according to the manufacturer’s guidelines. IncuCyte S3 (Sartorius, Göttingen, Germany) was used to quantify confluence and fluorescence (which was normalized using confluence). For measurements of mitochondrial superoxide levels, 50,000 cells per well seeded in flat-bottomed 24-well plates were stained with 5 μM MitoSOX red (Thermo Fisher) 30 min before treatment, according to the manufacturer’s guidelines, and harvested directly after treatment by trypsinization. MitoSOX fluorescence was quantified using a FACSCanto flow cytometer (BD Biosciences, Franklin Lakes, NJ, USA) and analyzed using FlowJo (version 10).

### 4.6. Quantification of Total ATP Levels

For total ATP quantification, 10,000 cells/well in flat-bottomed 96-well plates were treated with hyperthermia. ATP levels were assessed using Celltiter-Glo (Promega, Madison, WI, USA). Luminescence was quantified using a Clariostar plate reader. In parallel, the medium in additional wells was supplemented with 1 μg/mL Hoechst 33342 (Thermo Fisher), and ATP levels were normalized using Hoechst intensity to compensate for cell death.

### 4.7. Determination of Intracellular Copper Concentrations

Cells grown in 175 cm^2^ culture flasks (Greiner) were harvested 30 min after treatment by scraping, collected by centrifugation (5 min, 1200 rpm), and frozen in liquid nitrogen. The samples were thawed, mixed with double distilled 69% nitric acid and 1 mL of 30% hydrogen peroxide of Suprapur^®^ grade (Merck), and digested by closed-vessel microwave pressure digestion (ETHOS LEAN, Milestone), using the preprogrammed digestion program for animal products (35 min, 160–200 °C). After digestion, the samples were dried at 120 °C for two days and diluted in double-distilled 5% nitric acid. Copper concentrations were determined by ICP-OES using a Varian 720-ES (Varian Inc., Palo Alto, CA, USA).

### 4.8. Evaluation of Mitochondrial Membrane Potential

Cells were seeded in flat-bottomed 96-well plates at 10,000 cells/well. One hour before treatment, the medium was supplemented with 2 μg/mL JC1 (Thermo Fisher) for 20 min, followed by two washing steps with regular medium. The JC1 red and green intensity was read out according to the manufacturer’s guidelines, using a Clariostar plate reader.

### 4.9. Immunoblotting

Immunoblotting was performed and quantified as described previously [77], but the cells were harvested by trypsinization and lysed in Laemmli buffer containing complete proteasome inhibitor cocktail (Roche, Basel, Switzerland) and 10 mM N-ethylmaleimide (Selleck). Immunostaining for HSP70i, ubiquitin, and β-actin was performed with anti-HSP70i monoclonal mouse IgG (1:1000, Santa Cruz Biotechnology, Dallas, TX, USA), P4D1 monoclonal mouse IgG (1:500, Cell Signaling, Danvers, MS, USA), and anti-beta-actin polyclonal goat IgG (1:1000, Santa Cruz).

### 4.10. RNA Sequencing

Cells seeded in 10 cm dishes were treated as indicated and collected by scraping. RNA library preparation was performed using the RNeasy kit (Qiagen, Hilden, Germany), followed by a KAPA mRNA Hyperprep (Roche). RNA sequencing was performed on a NovaSeq 6000 in a 150 bp paired-ended fashion to a depth of 40 M reads. Reads were subsequently aligned in a paired-ended fashion using STAR (v2.7.10a), whereupon post-alignment processing was performed using SAMtools (v1.15.1) and unique molecular identifiers (UMIs) were demultiplexed with UMItools (v1.1.2). Finally, the mapped reads were assigned to genes using Subread (v2.0.1). Quality control was performed using FastQC (v.0.11.9) and MultiQC (v1.13). The steps were orchestrated using Snakemake (v7.14.0). Differential expression analysis was performed in R using DESeq2 [78]. Gene set enrichment and overrepresentation analyses were performed on shrunken log2 fold-changes by retrieving molecular signatures from the Molecular Signatures Database [79] with the msigdbr package, and testing for overrepresentation and enrichment was conducted using the clusterProfiler package. 

### 4.11. Fluorescence Microscopy and Image Analysis

Cells on coverslips were treated, washed with PBS, fixed with 2% paraformaldehyde (Thermo Fisher) in PBS, and permeabilized with PBS containing 0.1% Triton X-100 (Merck) and 1% FBS. Immunostaining for DLAT was performed in a permeabilization buffer with 1:100 dilution of mouse anti-DLAT (Cell Signaling) for 1 h, followed by washing with PBS and a 30 min incubation with 1:150 dilution of anti-mouse Cy3 (Jackson). Hoechst 33342 at 5 μg/mL was used to stain the cell nuclei, and fluorescence was protected using Vectashield (Vectorlabs, Newark, CA, USA). Z-stacks were captured using a Leica DM6i microscope equipped with a 40×/1.25/0.75 Plan Achromat oil objective (Leica, Wetzlar, Germany) and a Leica DFC9000 GT camera. Images were deconvolved using Huygens (SVI), and maximum-intensity projections were used to train a StarDist algorithm [80] with our own annotations, consisting of 50 patches, to detect DLAT-positive foci. The Cellpose “cyto2” algorithm [81] was used for nuclei segmentation to quantify the amount of DLAT-positive foci per nucleus.

### 4.12. Statistics

Statistical testing was conducted using Student’s *t*-test or two-way ANOVA with a Bonferroni post hoc test. The *p*-value threshold was 0.05. Error bars represent standard deviations. Boxplot whiskers represent a 5–95% confidence interval.

## 5. Conclusions

Hyperthermia has established a compelling clinical track record in treating diverse tumor types, and innovative approaches to heighten the thermosensitivity of cancer cells hold the potential to amplify its effectiveness. This study harnessed in silico methodologies to unveil novel strategies for enhancing thermosensitivity. Experimental endeavors undertaken to elucidate the underlying mechanism by which the copper ionophore elesclomol operates as a promising thermosensitizer have yielded a fundamental revelation: the disruption of copper homeostasis could serve as a broad approach to enhance the effectiveness of hyperthermia. While further mechanistic inquiries are imperative for a comprehensive understanding of how disruptions in copper homeostasis drive thermosensitization, we advocate for the initiation of translational investigations to assess the additional benefits of integrating elesclomol into localized hyperthermia-based treatments, such as bladder CHT, HILP, or HIPEC.

## Figures and Tables

**Figure 1 ijms-25-00423-f001:**
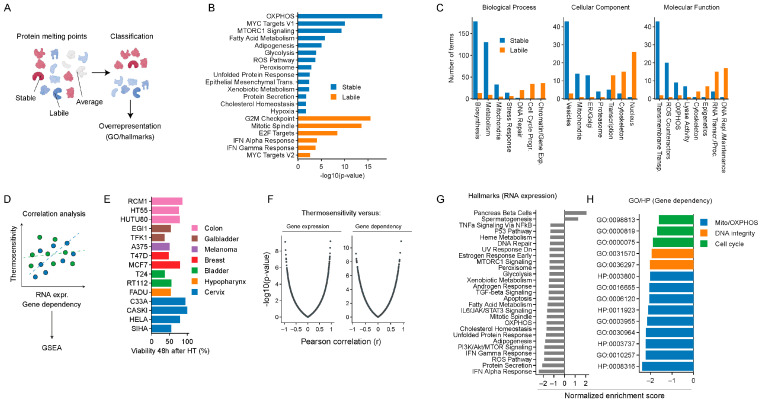
In silico analyses to pinpoint the potential therapeutic targets to promote thermosensitization: We categorized human proteins as thermostable or thermolabile, based on their melting point, as determined by Jarzab and colleagues [30], and overrepresentation analyses were carried out. (**A**) Analysis pipeline. (**B**) Overrepresented hallmarks in thermostable and -labile proteins. (**C**) Overrepresented Gene Ontology (GO) terms in thermostable and -labile proteins. We also investigated which genes contribute to cellular heat stress responses by performing two correlation analyses that utilized the thermosensitivity of various cancer cell lines. (**D**) Analysis overview. The thermosensitivity of the cell lines was correlated with RNA expression levels and CRISPR-inferred gene dependency. Genes were subsequently sorted based on Pearson correlation, and the result was used as the input for a gene set enrichment analysis (GSEA). (**E**) Thermosensitivity of a panel of cell lines, i.e., cell viability measured 48 h after exposure to hyperthermia (HT), categorized by tumor origin. (**F**) Global overview of correlation analysis results. (**G**,**H**) Enriched hallmarks, GO, and human phenotype (HP) terms in correlation-sorted genes.

**Figure 2 ijms-25-00423-f002:**
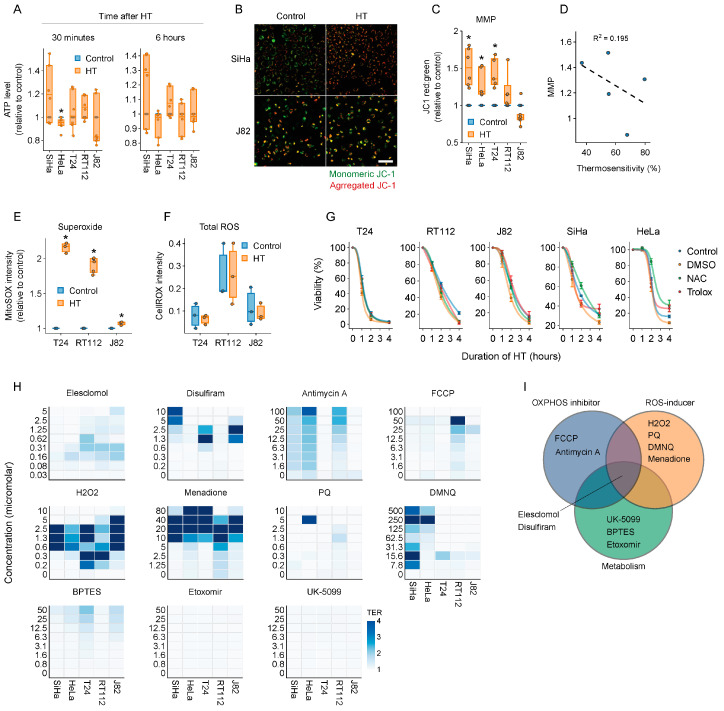
Discovery of novel thermosensitizing agents: (**A**) ATP levels 30 min and 6 h after exposure to hyperthermia (HT). (**B**) Representative images of JC1 staining, quantified in (**C**). Scale bar: 60 μm. (**C**) Changes in mitochondrial membrane potential (MMP), based on JC1 aggregation, measured directly after HT. (**D**) Pearson correlation between MMP, ATP levels, and thermosensitivity of five cell lines (T24, RT112, J82, SiHa, and HeLa). Next, to check whether the alteration of ROS homeostasis can lead to thermosensitization, we evaluated ROS levels after HT, as well as the effect of ROS scavengers on cell survival. (**E**,**F**) Superoxide and total ROS levels, measured directly after HT. (**G**) Cell viability 48 h after treatment with various thermal dosages, in the presence or absence of ROS scavengers. (**H**) Thermal enhancement ratios (TERs) of a selected drug panel, derived from cell viability assays performed 72 h after HT. (**I**) Categorization of the modes of action of the drug panel. Asterisks mark significant differences between the non-heated control and the hyperthermia arm.

**Figure 3 ijms-25-00423-f003:**
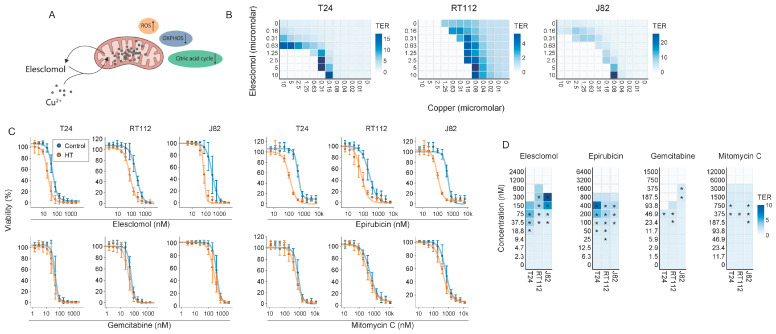
Elesclomol sensitizes to hyperthermia: We examined the extent to which elesclomol thermosensitizes in vitro. (**A**) Previously reported modes of action of elesclomol. (**B**) Thermal enhancement ratios (TERs) derived from cell viability assays performed 72 h after exposure to varying ratios of elesclomol and copper. TER values derived from cases where more than 90% cell killing was achieved by elesclomol monotherapy are not visualized, as they generally give rise to non-representative values. (**C**) Cell viability five days after exposure to elesclomol, mitomycin C, epirubicin, and gemcitabine. (**D**) TERs derived from (**C**). Asterisks mark significant differences between the non-heated control and the hyperthermia arm.

**Figure 4 ijms-25-00423-f004:**
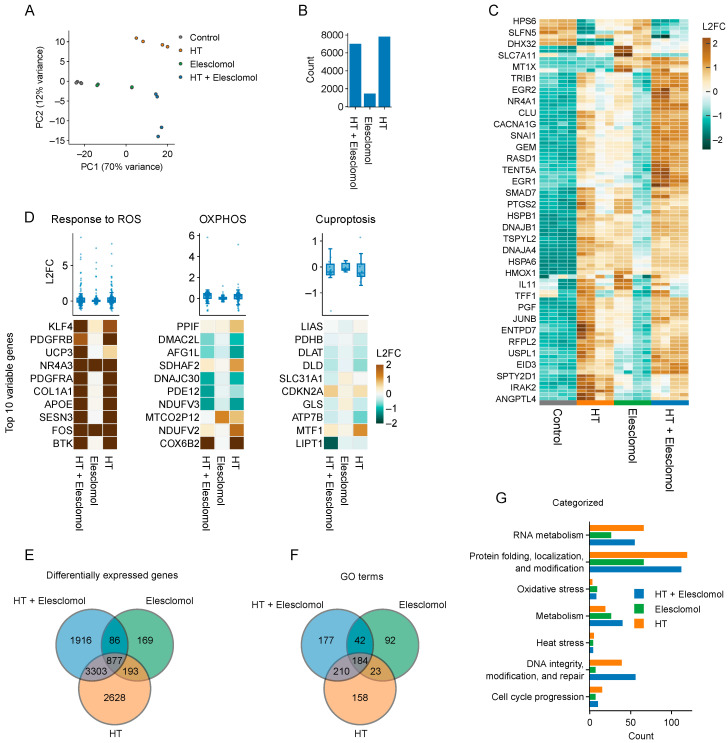
Transcriptomic alterations caused by combined treatment with elesclomol and hyperthermia: We treated cells with hyperthermia (HT), elesclomol (1 μM), or the combination and examined mRNA expression levels six hours after HT in T24 cells. (**A**) Principal component analysis of transcriptomes. (**B**) Numbers of significant differentially expressed genes per condition. (**C**) Heatmap showing the top 100 genes with the highest variation in expression among all conditions. Log 2 fold-change (L2FC) is centered and scaled row-wise. (**D**) Boxplots show the L2FC of all genes involved in cuproptosis (custom gene set acquired from Tang and colleagues [58]), oxidative phosphorylation (OXPHOS; GO:0006119), and responses to reactive oxygen species (ROS; GO:0000302) across all experimental conditions. Heatmaps show the L2FC of the top 10 variable genes. No scaling was applied to the L2FC. (**E**) Overlap between differentially expressed genes. (**F**) Overlap in Gene Ontology (GO) biological processes, as found by overrepresentation analysis. (**G**) Numbers of GO biological processes enriched per condition, subdivided into hand-curated categories.

**Figure 5 ijms-25-00423-f005:**
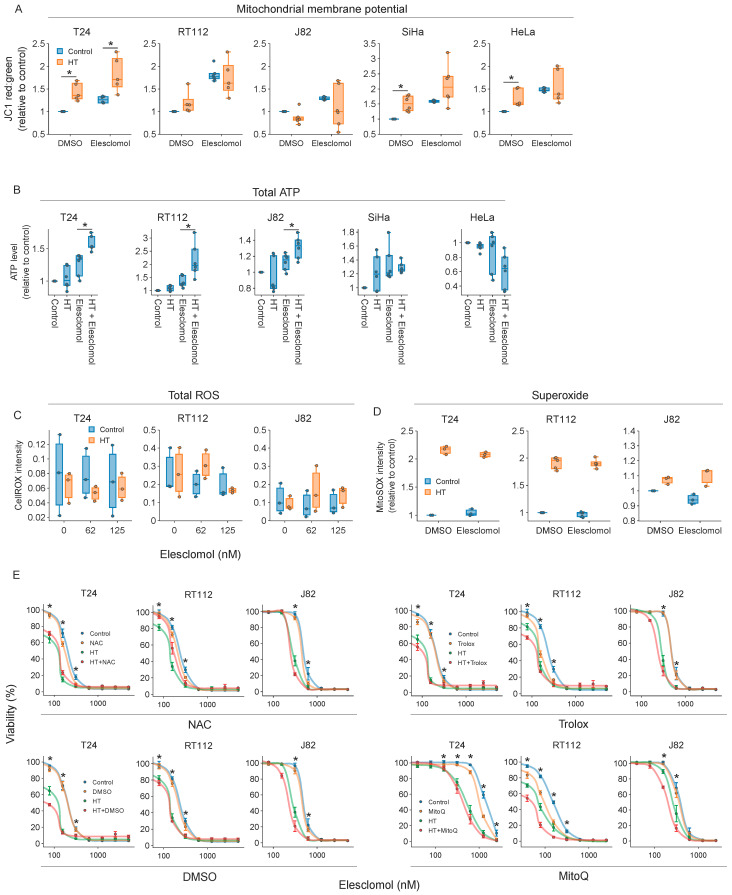
ROS induction and oxidative phosphorylation inhibition do not contribute to the thermosensitizing properties of elesclomol: (**A**–**C**) Mitochondrial membrane potential, total ROS, and superoxide levels directly after exposure to elesclomol (1 μM), hyperthermia (HT), or the combination. (**D**) Superoxide levels directly after exposure to the aforementioned treatments. Asterisks mark significant differences. (**E**) Cell viability 72 h after exposure to elesclomol (1 μM), HT, or both, in the presence/absence of ROS scavengers (NAC: 2 mM; Trolox: 1 mM; DMSO: 1%; MitoQ: 0.5 μM). Asterisks mark significant differences between the non-heated control and the hyperthermia arm.

**Figure 6 ijms-25-00423-f006:**
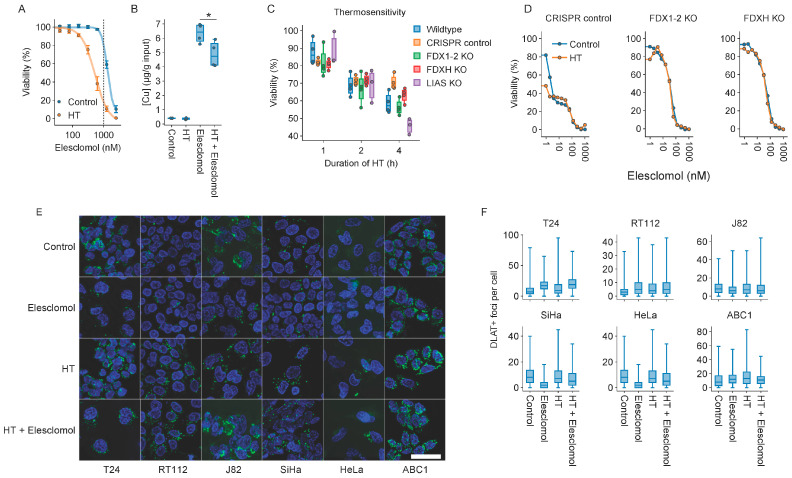
Thermosensitization by elesclomol occurs independently of cuproptosis: (**A**) Cell viability of T24 cells 72 h after treatment with hyperthermia (HT). The vertical line marks the elesclomol concentration of 1 μM. (**B**) Intracellular copper levels directly after HT, as determined by inductively coupled plasma optical emission spectrometry ICP-OES. (**C**) Thermosensitivity of ABC1 cells harboring a KO for ferredoxin 1 and 2 (FDX1-2), ferredoxin H (FDXH), or lipoic acid synthetase (LIAS), as determined by a cell viability assay, 72 h after exposure to HT. (**D**) Viability of wildtype and FDX1-2 and FDXH KO cells after treatment with elesclomol, HT, or a combination, 72 h after exposure to HT. (**E**) Representative images. Scale bar: 50 μm. Asterisks mark significant differences. (**F**) Number of DLAT-positive foci per cell 24 h after treatment with HT.

**Figure 7 ijms-25-00423-f007:**
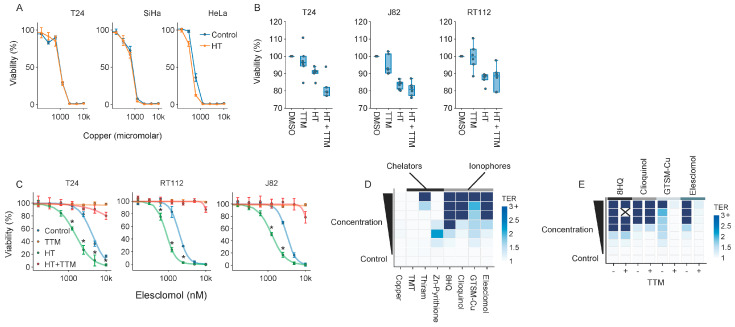
Elesclomol-induced thermosensitization is copper-dependent: (**A**) Cell viability 72 h after exposure to copper and/or hyperthermia (HT). (**B**) Cell viability 72 h after exposure to 10 μM of the copper chelator tetrathiomolybdate (TTM) and/or hyperthermia (HT). (**C**) Cell viability 72 h after exposure to elesclomol and/or hyperthermia (HT) in the presence/absence of 10 μM TTM. Asterisks mark significant differences between the non-heated control and the hyperthermia arm. (**D**) Thermal enhancement ratios (TERs) of a spectrum of copper and copper-binding compounds, derived from cell viability, assessed 72 h after HT. Cells were exposed to 1:2 dilution series, starting at 10 μM (copper, elesclomol), 150 μM (8HQ), 300 μM (clioquinol), 500 nM (GTSM-Cu), and 40 μM (thiram, TTM, Zn-pyrithione). TER > 3 is capped for improved visualization (**E**) TER of copper ionophores in the presence/absence of TTM, derived from cell viability assays, assessed 72 h after HT. TER values based on conditions where drug monotherapy caused >90% cell killing are removed (cross).

**Table 1 ijms-25-00423-t001:** Drug panel overview: Listing of the mode of action and the clinical developmental stage for drugs included in the panel. * DrugBank (accessed on 28 October 2022). GBM: glioblastoma multiforme; OXPHOS: oxidative phosphorylation; ROS: reactive oxygen species; H_2_O_2_: hydrogen peroxide; MPC: mitochondrial pyruvate carrier; CPT-1a: palmitoyltransferase 1a.

Compound	Category	Mode of Action	Clinical Phase *	Disease *
Etoxomir	Inhibitor of fatty acid oxidation	Blocks CPT-1a	-	-
UK-5099	Inhibitor of pyruvate catabolism	Inhibitor of MPC	-	-
BPTES	Inhibitor of glutamine catabolism	Inhibits glutaminase	-	-
FCCP	OXPHOS inhibitor	Complex V inhibitor	-	-
Antimycin A	OXPHOS inhibitor, ROS inducer	Complex III inhibitor	-	-
Menadione	ROS inducer	Redox cycling	I	Dermatologic neoplasms
DMNQ	ROS inducer	Redox cycling	-	-
PQ	ROS inducer	Redox cycling	-	-
H_2_O_2_	ROS inducer	Reacts directly with biomolecules	I/II	Breast cancer
Elesclomol	OXPHOS inhibitor, ROS inducer, cuproptosis inducer	Uncoupler of OXPHOS; transports copper to mitochondria	III	Melanoma, ovarian cancer, peritoneal cancer
Disulfiram			I/II	Melanoma, GBM, breast, prostate cancer, myeloma

## Data Availability

The data presented in this study are openly available in interactive form via figlinq.com (https://create.figlinq.com/dashboard/e.m.scutigliani:684, accessed on 22 November 2023).

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
