# Peer review of "Perturbation of Copper Homeostasis Sensitizes Cancer Cells to Elevated Temperature"

_ijms, 2023, doi:10.3390/ijms25010423_

Round 1
Reviewer 1 Report
Comments and Suggestions for Authors
The paper focuses on identification of sensitizers for heat treatment (hyperthermia) of tumors. In recent years, multimodal tumor therapies have made their way into clinics and it is of high scientific and clinical relevance to find new sensitizers for standard therapie such as radiotherapy, but also for complementary treatments such as hyperthermia (HT).
First, the authors follow an in silico approach to uncover compounds that could be used to thermo-sensitize cancer cells. They came up with elesclomol that is known as apoptosis inducer and to target cancer cell metabolism.
In vitro experiments with various cell lines suggest that elesclomol reduces the viability of several tumor cells similar to chemotherapeutics such as Gemcitabine. To what does the yellow and blue lines in Figure 3C refers to? The authors used the g PrestoBlue™ method for cell viability testing. Did they also analyze the induced cell death forms such as apoptosis and necrosis, as it is known from literature that elesclomol mainly induces apoptosis.
Next, the authors combined elesclomol with hyperthermia and revealed several transcriptomic changes. In vitro, ATP was increased after the combined treatments, but as shown in Figure 5 E, no significant thermo-sensitization of elesclomol was detected, at least with the PrestoBlue™ method. However, this can be seen in Figure 7C and here it was dependent on copper.
Generally, the introduction and discussion of the data is very short, while many results are presented. Thereby, the reader gets a bit lost and sometimes contradictory findings are shown (e.g. Figure 3C and 7C regarding thermo-sensitization of elesclomol). There is no doubt that the findings are new and valuable, but they should be presented in a more concise manner.
Reviewer 2 Report
Comments and Suggestions for Authors
The submitted study investigates the role of copper in hyperthermia. By compound screening, the authors identified elesclomol as potential sensitizer. They showed that elesclomol action was independent of its previously reported modes of action, but depended on copper shuttling.
Comments
-The authors used a broad panel of cell lines for the first screening but in the later studies used mainly cervix carcinoma and bladder cell lines. It might be helpful to explain this by the better accessibility of hyperthermia treatment of the respective cancers.
-The cell lines used in the focused screening reacted differently to elesclomol. It would be helpful to know, if the effects can be linked to any physiological parameter (antioxidant levels, proliferation rate, glucose metabolism, etc.)
-What would be the proposed administration of elesclomol? Combination with copper as used in the study? What about the systemic toxicity of copper?
-The authors found that the commonly known modes of elesclomol action were not involved in the sensitizing effect – do they have a suggestion of the potential mechanism of action?
Minor
Culture conditions of the ABC-1 cells was not described.
Immunoblotting is described but no data presented.
Journal abbreviations are not consistent.
Round 2
Reviewer 1 Report
Comments and Suggestions for Authors
The authors adequately revised the mansucript.